# Recent Advances in the Synthesis and Applications of m-Aryloxy Phenols

**DOI:** 10.3390/molecules28062657

**Published:** 2023-03-15

**Authors:** Dinara Amankulova, Gulzat Berganayeva, Bates Kudaibergenova, Dinara Zhetpisbay, Ayshagul Sharipova, Moldyr Dyusebaeva

**Affiliations:** 1Faculty of Chemistry and Chemical Technology, Al-Farabi Kazakh National University, 71 al-Farabi Ave., 050042 Almaty, Kazakhstan; 2Department of Biochemistry, Asfendiyarov Kazakh National Medical University, 94 Tole bi Ave., 050012 Almaty, Kazakhstan; 3Chemical and Technological Faculty, Karakalpak Berdah State University, 1 Ch. Abdirov St., Nukus 230100, Uzbekistan

**Keywords:** m-aryloxy phenols, potential biological activities, hydroxylation of benzenes, nucleophilic aromatic substitutions, electrophilic aromatic substitutions, metal-catalyzed cross-coupling reaction

## Abstract

Since phenol derivatives have high potential as building blocks for the synthesis of bioactive natural products and conducting polymers, many synthesis methods have been invented. In recent years, innovative synthetic methods have been developed for the preparation of m-aryloxy phenols, which has allowed for the preparation of complex m-aryloxy phenols with functional groups, such as esters, nitriles, and halogens, that impart specific properties of these compounds. This review provides an overview of recent advances in synthetic strategies for m-aryloxy phenols and their potential biological activities. This paper highlights the importance of m-aryloxy phenols in various industries, including plastics, adhesives, and coatings, and it discusses their applications as antioxidants, ultraviolet absorbers, and flame retardants.

## 1. Introduction

m-Aryloxy phenols have a wide range of applications, including as antioxidants, ultraviolet absorbers, and flame retardants. They are commonly used in the production of plastics, adhesives, and coatings due to their ability to improve these materials’ thermal stability and flame resistance. In addition, m-aryloxy phenols have been found to have potential biological activities, including anti-tumor and anti-inflammatory effects [1].

Phenol derivatives have been widely researched, as they have high potential for synthesizing bioactive natural products and conducting polymers. Many synthesis methods have been developed for phenol derivatives, with conventional strategies focusing on functionalizing and transforming functional groups around the aromatic ring. These strategies include selective hydroxylation of benzenes, nucleophilic and electrophilic aromatic substitutions, and metal-catalyzed cross-coupling reactions [2].

The most successful synthesis strategies were developed based on the Ullmann copper-mediated aromatic nucleophilic substitution reactions, which were reported in the early 1900s by Fritz Ullmann and Irma Goldberg. In the original Ullmann reaction protocol, stoichiometric amounts of copper salt were used, along with high reaction temperatures (≥200 °C) and prolonged reaction times. However, the modern variants of the Ullmann reaction have focused on using copper-mediated (stoichiometric or catalytic) reactions between aryl halides and other reagents, such as amines, phenols, or thiophenols, to produce aryl-amine, -ether, or -thioether compounds, respectively (Figure 1) [3,4].

Despite the expanded scope of the Ullmann reaction, the term primarily refers to the copper-catalyzed synthesis of biaryls from aryl halides. This protocol has been widely adopted due to its high efficiency and versatility in constructing complex biaryl compounds. The Ullmann reaction remains an essential tool in modern organic synthesis and has contributed significantly to the development of many natural products, pharmaceuticals, and materials [5].

Although nucleophilic aromatic substitution is a highly effective method for synthesizing phenol derivatives, the strong influence of the hydroxyl group on ortho/para substitution makes it difficult to prepare meta-substituted phenol derivatives.

The synthesis of m-aryloxy phenols is a significant area of research, as it allows for the preparation of new compounds with tailored properties and applications. In light of their unique properties and broad range of applications, m-aryloxy phenols represent an important area of research with significant potential for future developments. This paper reviews the recent advances in the synthesis of m-aryloxy phenols, as well as their applications in various industries, and highlights the potential for further research in this field.

## 2. Synthesis of m-Aryloxy Phenols by Demethylation of m-Methoxy Phenols

The demethylation reaction involves the elimination of a methyl group from a molecule. One example of this reaction is the demethylation of m-methoxy phenols, which forms m-aryloxy phenols.

There exist multiple approaches for demethylation, including chemical and catalytic methods. The most prevalent chemical method involves the utilization of potent acids, such as sulfuric acid, hydrochloric acid, or nitric acid. These acids convert the methoxy group into a hydroxyl group [6].

In contrast, catalytic demethylation methods usually use transition metal catalysts, including copper or palladium. These methods can be employed through various mechanisms, including hydrogenation, transfer hydrogenation, and activation of C-H bonds.

The use of hydrogen bromide and boron tribromide (BBr_3_) as catalysts in demethylation reactions represents a valuable approach for the synthesis of m-aryloxy phenols. Bronsted acid HBr and Lewis acid BBr_3_ can coordinate with electron-rich sites in organic compounds and enhance the outcome of organic reactions. This property enables the efficient demethylation of methoxy phenols to m-aryloxy phenols [7,8].

In Kormos and his team’s research, 4-(3-hydroxyphenoxy) benzoic acid was synthesized from 4-(3-methoxyphenoxy) benzoic acid by refluxing it with 48% hydrogen bromide in acetic acid. By condensing the 4-(3-hydroxyphenoxy) benzoic acid with piperazine using N-ethylcarbodiimide hydrochloride (EDC·HCl) and a catalytic hydroxy benzotriazole (HOBt), the team synthesized N-(1S)-1-{[(3S)-4-(3-hydroxyphenyl)-3-methylpiperazin-1-yl]-methyl}-2-methylpropyl-4-(3-hydroxyphenoxy) benzamide (Figure 2) [9].

This compound showed strong antagonistic properties at the κ opioid receptor, with inhibition agonist-stimulated [35S]GTPγS binding in cloned human μ (DAMGO) -Ke(nM)a = 69 ± 14, δ (DPDPE) − Ke(nM)a = 625 ± 120, and κ (U69,593) − Ke(nM)a = 1.85 ± 0.51. The authors believed these compounds will be useful in the development of clinical candidates for treating conditions such as depression, anxiety, schizophrenia, and addictions, as well as serving as pharmacological tools [9,10,11].

In 2022, Yamamoto et al. reported the synthesis of N-(2-(3-hydroxyphenoxy)-4-nitrophenyl)methanesulfonamide through the demethylation of the corresponding methoxy derivatives by utilizing BBr_3_ as a dealkylating agent (Figure 3). The resulting compound was utilized to synthesize isomeric [^11^C] methoxy analogs of nimesulide as a suitable radiotracer candidate for imaging the expression of Cyclooxygenase-2 (COX-2) in the brain. COX-2 expression has been documented in various tumor types, including head and neck cancers, colorectal cancers, and melanomas [12,13].

In 2009, Yang [14] obtained 3,3′-oxydiphenol from the interaction of 3-methoxyphenol with 3-bromanisole in the presence of a Cu-catalyst and subsequent cleavage of methyl groups with HBr in acetic acid. In [15], 3,3′-oxidiphenol was used as a diaryl building block to produce various oxacalix[*n*]arenes with an odd number of aromatic units (*n* = 5, 7) in high yields. Calixarenes have hydrophobic cavities that can hold smaller molecules or ions, and they belong to a class of cavitands known in host–guest chemistry. Thanks to selective sorption, calixarenes have been widely used in practice, for example, as receptors for selective or group extraction of ions of various metals [16,17,18], catalysts, and enzymes [19,20,21].

## 3. Synthesis of Aryloxy Phenols by Reactions between Aryl Halides and Resorcinol

The reaction between aryl halides and resorcinol involves a nucleophilic aromatic substitution mechanism. In this mechanism, the resorcinol undergoes deprotonation under basic conditions and, subsequently, acts as a nucleophile on the aromatic ring of the aryl halide. This results in the formation of an intermediate species, in which the nucleophile attacks the ring, leading to the substitution of the halide group. The intermediate species contains an oxide anion, which is formed due to the loss of a proton. The final product of the reaction is an aryloxy phenol, which can be obtained in a good quantity under appropriate reaction conditions [22].

In 2010, Vagin and colleagues published a study detailing the synthesis of 5-(3-hydroxyphenoxy)-2-nitroaniline from 5-chloro-2-nitroaniline and resorcinol. The reaction was performed using sodium hydride as a base and DMF as a solvent and was carried out under an argon atmosphere by heating the reaction mixture at 125 °C for 24 h. They then used a hydrogenation reaction with a Pd/C catalyst to obtain 4-(3-hydroxyphenoxy)-1,2-benzyldiamine (Figure 4) [23].

The resulting compound was employed in the development of a reliable and versatile method for creating dinuclear salphen systems [24]. The use of corresponding Cr (III) complexes demonstrated increased activity in significant reactions, such as the polymerization of γ-butyrolactone and the copolymerization of CO_2_ with propylene oxide, highlighting the importance of bimetallic processes [23].

In 2015, Wang and his team prepared a self-promoted hydroxy-containing phthalonitrile system, comprised of 1,3-bis(3,4-dicyanophenoxy) benzene (BDB) and 4-(3-hydroxyphenoxy) phthalonitrile (HPPH) through an in situ reaction in a one-pot nucleophilic displacement reaction. This reaction involved the removal of a nitro-substituent from 4-nitrophthalonitrile (Figure 5) [25].

The resulting HPBD material was found to possess desirable processability and mechanical properties, including a high modulus, a high glass transition temperature, and excellent thermal stability. The polymerization of HPBD resulted in the formation of a phthalocyanine ring, an isoindole ring, and a triazine ring. Additionally, post-curing at high temperatures was observed to improve thermal stability and produce void-free resin structures, making HPBD suitable for use in industries such as aerospace and microelectronics [25,26].

In 2011, Bollini et al. published a synthesis of 4-(3-hydroxyphenoxy)-3-((tetrahydro-2H-pyran-2-yl)oxy) benzonitrile and 4-(3-hydroxyphenoxy)-3-benzyloxybenzonitrile from 4-fluoro-3-hydroxybenzonitrile through a two-step process (Figure 6) [27].

The resulting compounds served as intermediates in the investigation of novel non-nucleoside reverse transcriptase inhibitors (NNRTIs) for the treatment of HIV [28]. The final compound, possessing a methoxy–ethoxy substituent, was obtained in four steps from 4-(3-hydroxyphenoxy)-3-((tetrahydro-2H-pyran-2-yl)oxy) benzonitrile and demonstrated significant activity, with an inhibitory concentration of 0.54 μM [27,29,30].

In 2011, Xia and colleagues developed a novel class of benzoxaborole β-lactamase inhibitors through a three-step synthesis process. First, they carried out a nucleophilic substitution of 2-bromo-4-fluorobenzaldehyde with resorcinol to obtain an ether. Then they used palladium-mediated boronylation on the ether to produce an aldehyde, which was finally reduced with NaBH_4_ and underwent acid-catalyzed cyclization to create the final product—3-hydroxyphenoxybenzoxaborole (Figure 7) [31,32].

The 3-hydroxyphenoxybenzoxaborole was found to be 2–3 times more potent than benzoxaborole in inhibiting CTX-M-9a [31].

In 2016, Kobayashi and his team synthesized 5-chloro-4-[4-chloro-2-fluoro-5-(3-hydroxyphenoxy-phenyl]-1,2-tetramethylene-4pyrazolin-3-one from 2,5-difluoro-4-bromonitrobenzene in seven steps, using the nucleophilic aromatic substitution reaction between 2,5-difluoro-4-bromonitrobenzene and 3-methoxy phenol to obtain an ether. The elimination of a methyl group using boron tribromide as a demethylation agent in the final step produced 5-chloro-4-[4-chloro-2-fluoro-5-(3-hydroxyphenoxy)-phenyl]-1,2-tetramethylene-4-pyrazolin-3-one (Figure 8) [33].

The resulting compound showed promising results as an herbicide, with a good control effect on weeds, good sustainability, and selective action between crops and weeds. It can be used effectively as an ingredient in herbicides [33,34].

In 2018, Kazuia et al. successfully synthesized 3-(4-aminophenoxy)-phenol from p-fluoronitrobenzene and resorcinol (Figure 9). The reaction was conducted using sodium hydroxide and DMSO at 50 °C for three hours. The final product was obtained after hydrogenation using a Pd/C catalyst in methanol and a hydrogen atmosphere for two hours [35].

The resulting compound was used to create a series of (benzoylaminophenoxy)-phenol derivatives that showed promising activity as anti-prostate cancer agents [35,36,37].

In 2018, Frączk and his team published a synthesis of 2-(3-hydroxyphenoxy)-5-chlorobenzonitrile in two steps. The process involved a reaction of 3-fluoro-5-chlorobenzonitrile with 3-methoxyphenol in NMP and potassium carbonate in the first step, followed by demethylation using BBr_3_ in DCM in the second step (Figure 10) [38].

The compound resulting from the synthesis by Frączk and his group in 2018 was an intermediate in creating novel diaryl ethers, which are NNRTIs with improved solubility. These inhibitors showed promising results with IC_50_ values at low micromolar to sub-micromolar concentrations [38,39,40].

In 2022, a group led by Zhong published a synthesis method for producing 3-[2-chloro-4-(trifluoromethyl)phenoxy]phenol. The synthesis was accomplished through the displacement of 2-chloro-1-fluoro-4-(trifluoromethyl)benzene with resorcinol and required a high temperature (130 °C) and an inert atmosphere for a 24-h reaction time (Figure 11) [41].

3-[2-chloro-4-(trifluoromethyl)phenoxy]phenol is an intermediate in the synthesis of MK-2305, a highly potent agonist for the G-protein-coupled receptor 40 [41,42].

The utilization of a catalyst in the nucleophilic aromatic substitution reaction between an aryl halide and resorcinol has been widely investigated. The selection of a suitable catalyst is contingent upon the reaction conditions and the desired outcome. Transition metal catalysts, including palladium and copper, have been demonstrated to be effective in catalyzing this reaction, potentially leading to an enhancement in reaction rate and/or improved regioselectivity. However, the implementation of a catalyst may also introduce practical drawbacks, such as an increased cost and the requirement for specific handling and disposal procedures. Copper chloride (CuCl) has been proven to be an effective catalyst in the reaction between an aryl halide and resorcinol, resulting in an improvement in reaction rate and/or regioselectivity [43,44,45,46].

In 2010, Xue and colleagues synthesized m-aryloxy phenols through a two-step procedure. First, they performed Ullmann couplings of m-methoxy phenol with iodobenzene derivatives using Cs_2_CO_3_ and a small amount of CuBr as a catalyst to produce the intermediate compound. Then they cleaved the methyl ethers using BBr_3_, resulting in m-aryloxy phenols with high yields. These m-aryloxy phenols were then used to synthesize a new series of inhibitors of nNOSs, which were evaluated for their ability to inhibit three different NOS isozymes: rat nNOSs, bovine eNOSs, and murine iNOSs (Figure 12) [47,48].

In 2010, Sapkota and colleagues published a study on a two-step copper catalyst-based synthesis of eight meta-aryloxy phenols from methoxy phenols and 2-bromonitrobenzene or 2,4-dibromo nitrobenzene. The yields of the final compounds varied between 7 and 58% (Figure 13) [49].

The synthesized compounds were evaluated for their anti-oxidative properties by measuring their ability to reduce the DPPH free radical to DPPH-H. However, the majority of the compounds were inactive in this regard, with only a few compounds possessing a resorcinol structure exhibiting moderate antioxidant activity, although it was lower than that of ascorbic acid [49,50].

In 2010, Bouey and her team synthesized 3-(p-tolyloxy)phenol from 3-iodoanisole and p-methylphenol through a two-step process (Figure 14). The first step involved etherification utilizing copper iodide, N,N-dimethylglycine hydrochloride, and cesium carbonate. The second step involved demethylation through the use of boron tribromide and DCM as solvents [51].

The 3-(p-tolyloxy)phenol was then used to synthesize polysubstituted imidazolone derivatives, which possess peroxisome proliferator-activated receptor (PPAR) agonist properties and can be used for the treatment of certain conditions related to lipid and glucose metabolism disorders, as well as hypertension. These conditions include metabolic syndrome, diabetes, dyslipidemias (abnormal levels of lipids in the blood), and obesity, which can all increase the risk of developing cardiovascular diseases [51,52].

Lee and his team reported a method for the synthesis of 3-(4-bromophenoxy)phenol, a meta-substituted phenol derivative, using the Ullmann coupling reaction (Figure 15). The reaction involved the use of copper iodide, potassium carbonate, and L-proline as catalysts and was conducted in DMSO [53].

The synthesized 3-(4-bromophenoxy)phenol was then utilized as a component in the fabrication of an organic light-emitting device (OLED) [54]. OLEDs consist of two electrodes and an organic layer that contains at least one organometallic compound, and they are known for their self-emissive properties that result in full-color images with high brightness, high contrast ratios, and wide viewing angles [53].

In 2012, Li and his team reported a two-step synthesis of bisphenol monomers from 4,4′-dibromo biphenyl and 3-methoxyphenol using a copper catalyst (Figure 16). The intermediate methoxy-terminated four-ring monomer was hydrolyzed using hydrobromic acid in acetic acid to produce alcohols. Both steps required high-temperature conditions of 165 °C for the first step and 150 °C for the second [55]. 

The resulting bisphenol monomers were used to prepare novel ionic aromatic polymers with precisely sequenced ionic moieties that contained one or two sulfonic acid groups along the backbone. Sulfonated aromatic hydrocarbon polymers have garnered attention as potential candidates for membranes in the production of electrical power and clean water [55,56].

In 2013, Bartholomeus et al. published a report on the synthesis of 3-(3,5-dimethoxyphenoxy)phenol using the Ullmann-type coupling reaction (Figure 17). The process involved combining 1-bromo-3,5-dimethoxybenzene with resorcinol in the presence of a copper iodide catalyst, N, N-dimethylglycine hydrochloride, cesium carbonate, and DMF as the solvent [57].

The resulting product obtained from the synthesis process was an intermediate in the creation of fulicineroside. This compound demonstrated moderate inhibition against Staphylococcus aureus and Bacillus subtilis. Additionally, it was found to have significant inhibitory effects on crown gall tumors, suggesting its potential as an anti-tumor agent [57,58].

The authors of [59] describe a method for obtaining 1,3-di(3-hydroxyphenoxy)benzene which includes two stages. This method is a modified form of the process published by Wang [60]. Thus, at the first stage, 1,3-diaryl-substituted products are produced through the interaction of various activated m-cresols with protected hydroxy groups with compounds containing leaving groups (LG) in the meta position in the presence of a catalyst such as copper (I) chloride. Further interaction of these compounds with an acid catalyst leads to the formation of 1,3-di(3-hydroxyphenoxy)benzene (Figure 18).

Protecting groups, which can be temporary or permanent, are known by their art, and methods for their installation and removal are described in standard references [61], the contents of which are incorporated by reference. Hydroxyl groups can be protected using groups (P) such as acetyl, benzoyl, benzyl, methoxymethyl, methoxytrityl, methylthiomethyl, pivaloyl, tetrahydropyranyl, tetrahydrofuran, trityl, silyl (including trimethylsilyl, tert-butyldimethylsilyl, tri-isopropylsilyloxymethyl, and triisopropylsilyl), alkyl (such as methyl ethers), and ethoxyethyl.

1,3-Di(3-hydroxyphenoxy)benzene is the basis of epoxy resins with high resistance to deformation. Epoxy resins are versatile materials that can be combined with fibers to produce a variety of composite materials, including a raft of prepreg compositions.

In 2015, Gim et al. published a method for synthesizing 3-(p-substituted aryloxy) phenols through an Ullmann coupling reaction. This reaction involved combining resorcinol with aryl iodides using CuI and picolinic acid as catalysts (Figure 19). The resulting compounds were then utilized to develop a new series of alkoxy-3-indolylacetic acid analogs, which showed potential as PPAR agonists. These compounds have been tested for treating dyslipidemia, obesity, and insulin resistance in clinical trials [62,63].

In 2016, Yang and his colleagues introduced a four-step method for synthesizing 3-(3,5-dichloro-4-hydroxyphenoxy)phenol from 4-iodophenol (Figure 20). The process involves the Ullmann reaction catalyzed by copper and using 3-methoxyphenol as a co-reagent, followed by demethylation with HBr in acetic acid. All steps require an inert atmosphere and exhibit high yields. The same procedure was also used to synthesize 3-(3,5-dichloro-4-hydroxyphenoxy)-4-chlorophenol. This process was described by Hu and his group in 2016 [64,65].

The resulting compound was utilized to create compounds aimed at enhancing fluorogenic probes for measuring, detecting, and screening hypochlorous acid or hydroxyl radicals in biological and chemical samples, such as cells and tissues within living organisms [63].

In 2017, Bai and his team published a two-step synthesis of 3-(3,5-diiodo-4-hydroxyphenoxy)phenol and 3-(3,5-diiodo-4-hydroxyphenoxy)-4-chlorophenol from 2,4,6-triiodophenol with corresponding reagents. The final step involved using boron tribromide in DCM as a demethylation agent (Figure 21) [66].

The resulting compounds were utilized to create a highly sensitive and selective probe, HKOH-1, which had an ability to monitor hydroxyl (-OH) formation in live cells that was improved by >30-fold [66,67].

According to the patent by Breitenburcher and colleagues in 2010 [68], the desired compounds were synthesized by treatment of the appropriate starting material with a CuI solution in the presence of resorcinol, dimethylglycine HCl, and cesium carbonate in N,N-dimethylacetamide (DMA) (Figure 22).

The resulting products were utilized as intermediate reagents in the synthesis of aryl-substituted heterocyclic urea derivatives, which possessed the ability to modulate the activity of fatty acid amide hydrolase (FAAH). The authors described methods of utilizing these aryl-substituted heterocyclic urea derivatives in the treatment of various disease states and disorders [68].

## 4. Sonogashira Coupling: A Copper-Catalyzed Method for Biaryl Synthesis

The Sonogashira coupling reaction represents a variation of cross-coupling, wherein the interaction between aryl boronic acids and phenols is facilitated by the presence of a copper catalyst and a base. Copper (II) acetate (Cu(OAc)_2_) serves as a typical example of a copper catalyst for this reaction, while potassium carbonate (K_2_CO_3_) is commonly used as the base. The reaction mechanism involves the formation of a complex between the copper and the phenol, which subsequently reacts with the aryl boronic acid to form an intermediate. This intermediate undergoes transmetallation to furnish the final biaryl product. Due to its high efficiency and versatility, the Sonogashira coupling reaction has gained widespread use in the synthesis of pharmaceuticals and agrochemicals, among other applications [69,70].

A report by Bryan and colleagues in 2015 summarizes research endeavors centered on forming new carbon-heteroatom bonds using organoboron reagents through copper acetate-mediated reactions under ultrasound irradiation (Figure 23). The methodology involved incorporating ultrasound irradiation in the Chan–Evans–Lam reaction to achieve the O-arylation of phenols, the N-arylation of anilines and indoles, and the S-arylation of thiols. The utilization of ultrasound irradiation was discovered to significantly reduce the reaction times from 72 h to 4 h while increasing product yields by an average of 20% [71].

Scientists at the University of Pisa described a method for producing new diaryl ether phenolic compounds, which are two peripheral phenolic rings with hydroxyl groups in the meta- and/or para-position, which are connected by a central 1,3-disubstituted phenyl ring. The synthesis of these compounds included four stages (Figure 24).

The synthesis was started with commercially available 3-bromophenol, which is subjected to a cross-coupling reaction under bis(triphenylphosphine)palladium-catalyzed Suzuki conditions with 3- or 4-methoxybenzene boronic acid to obtain diaryl derivatives—3- and 4-phenoxy anisole in quantitative yield (99%) (Figure 24, stage 1). The catalyst was obtained by mixing a solution of palladium acetate Pd(OAc)_2_ and triphenylphosphine (PPh_3_) in ethanol and toluene at room temperature in an inert gas environment. The formation of a C–O bond interaction of these intermediates with 3- and 4-bromanisole was carried out in the presence of appropriate catalysts during heating. In fact, they reacted in a ligand-free Ullmann-type reaction with 4-bromoanisole and forms 3- and 4-methoxy substituted derivatives of diaryl ether (Figure 24, stages 2 and 3). All the methoxy-substituted derivatives were subjected to a common last synthetic step, consisting of a BBr_3_-promoted removal of the methyl groups, in order to obtain the desired diaryl ether phenolic compounds (Figure 24, stage 4). The use of this dealkylating agent provided good yields of the target products (yield 71–95%) [72].

The synthesized diaryl ether phenolic compounds were able to activate the human enzyme SIRT1 and regulate many metabolic functions. They can be used for medical applications, specifically for the treatment or prevention of cardiometabolic diseases such as diabetes, heart failure, and atherosclerosis.

## 5. Synthesis of m-Aryloxy Phenols Using Grignard Reagents

The utilization of Grignard reagents, which are a type of organometallic compound, in the synthesis of aryloxy phenols via a reaction with aldehydes or ketones has received widespread attention in the field of organic chemistry. Through interaction with a carbonyl compound, a complex intermediate is formed, which, upon treatment with acidic conditions, gives rise to the corresponding alcohol. The alcohol can then be oxidized through the use of oxidation reagents such as hydrogen peroxide (H_2_O_2_) or sodium hypochlorite (NaClO) to produce the desired aryloxy phenol.

It is important to note that the reaction conditions, such as the choice of solvent and the presence of catalysts or bases, have a significant impact on the reactivity, selectivity, yield, and stereochemistry of the Grignard reaction, as well as the final product [73].

In 2012, researchers led by Pidathala used 4-(3-methoxyphenoxy)benzaldehyde in a Grignard reaction to produce an intermediate alcohol (Figure 25).

The alcohol was then oxidized using PCC to obtain 1-(4-(3-methoxyphenoxy)phenyl)propane-1-one with a yield of 89%. BBr_3_ was utilized to remove the methyl and produce 1-[4-(3-hydroxyphenoxy)phenyl]propane-1-one with a yield of 62% [74].

The resulting compound was used to synthesize a series of bisaryl quinolones with potent antimalarial activity. The lead compounds in this series showed antimalarial activity against the 3D7 strain of P. falciparum and had low nanomolar PfNDH2 activity [74,75].

Gao and colleagues synthesized 3-(4-fluorophenoxy)phenol from arylmetals via hydroxylation using N-benzyl oxaziridine. The reaction proceeded at room temperature for a duration of 2 h. The authors demonstrated that bench-stable N−H and N−alkyl oxaziridines, derived from economical terpenoid scaffolds, can act as effective primary aminating and hydroxylating reagents, producing primary arylamines and phenols directly from readily available aryl-metals in the absence of transition-metal catalysts and under exceptionally mild conditions (Figure 26) [76].

## 6. Hydrolysis of Diazonium Salts Using a Two-Phase System

In 2015, Taniguchi et al. published findings on the synthesis of 3-phenoxyphenol and 3-(4-nitrophenoxy)phenol (Figure 27). The method involved the hydrolysis of intermediate diazonium salts derived from anilines in a two-phase system of cyclopentyl methyl ether (CPME) and water, resulting in high yields of the desired compounds.

These products were found to be key components for the production of raw materials for functional plastics, specifically polyimide resin [77].

## 7. Synthesis of 3-Aryloxyphenols from 3-Chlorocyclohex-2-en-1-one

In 2023, a chemical reaction was reported by Clive et al. wherein 3-chlorocyclohex-2-en-1-one was treated with phenols in the presence of K_2_CO_3_, resulting in the formation of 3-(aryloxy)cyclohex-2-en-1-ones. The synthesized products were then subjected to bromination at the C(2) position and using NBS in DMF, which ultimately formed the brominated products. Subsequently, the brominated compounds were aromatized to form 3-(aryloxy)phenols through a treatment with DBU in PhMe or MeCN. This multi-step reaction sequence necessitated the use of several reagents and conditions to generate the final compounds (Figure 28). The method presently utilized operates at ambient temperature and does not employ any heavy metals or ligands. Moreover, it circumvents the procedural steps required to overcome the o,p-directing effect of oxygen, owing to the readily available 1,3-functional group relationship inherent in the starting material, namely cyclohexane-1,3-dione [78].

## 8. Conclusions

In summary, the past decade has born witness to remarkable advances in the synthesis and utilization of m-aryloxy phenols. The implementation of various synthetic methodologies has resulted in the preparation of structurally diverse m-aryloxy phenols, including biologically active compounds with promising pharmacological applications. Moreover, m-aryloxy phenols have demonstrated versatility in the fields of materials science and organic electronics. These achievements have propelled further research on the synthesis, characterization, and application of m-aryloxy phenols, and it is anticipated that this research area will continue to expand in the foreseeable future. The growing interest in m-aryloxy phenols underscores their potential to emerge as key compounds in the field of synthetic organic chemistry and their relevance to a diverse range of technological applications.

## Data Availability

Not applicable.

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
