# Peer review of "Recent Advances in the Synthesis and Applications of m-Aryloxy Phenols"

_molecules, 2023, doi:10.3390/molecules28062657_

Round 1

Reviewer 1 Report

I carefully read this review several times and find it is very good. The contents were also in great details. I think it does not need special modifications and can be accepted in its current form.

Author Response

Dear Reviewer,

Thank you very much for your thoughtful review of our manuscript. Your positive feedback is much appreciated, and we are glad that our efforts have been able to meet your expectations. We would like to express our gratitude for your time and effort in evaluating our work and for your positive recommendation for its acceptance.

We hope that our manuscript will be able to contribute to the scientific community and be of value to researchers and readers in this field. Once again, thank you for your review, and we look forward to hearing from the journal regarding the publication status of our manuscript.

Sincerely,

Moldyr Dyusebaeva

Reviewer 2 Report

The authors present a review article on recent advances on the synthesis of m-aryloxy phenols and their applications.  The article is well-written and good examples of both synthesis and applications are provided.  However, prior to acceptance, some changes should be considered:

1.  Percent yields should be consistently included in schemes.  Most have percent yields included, but many (e.g., 5, 7, 10, etc.) do not.

2.  The structure of resorcinol should be provided in the first reaction scheme (Scheme 4) in which it is mentioned.  Not all readers will be immediately familiar with the structure. 

3.  Line 121:  this is surely NOT an electrophilic aromatic process.  It is a nucleophilic aromatic substitution in which the resorcinol is deprotonated under the basic reaction conditions to act as the nucleophile on the aromatic ring containing electron-withdrawing groups.  Furthermore (as per line 124), there is NOT an an oxonium ion (positively charged) intermediate involved in the reaction.  It would be an oxide (negatively charged) intermediate.

4.  Line 129:  an "o"and "e" missing from 5-chloro-2-nitroaniline

5.  Scheme 6:  the reagent should be PhCH2Br (i.e., benzyl bromide) and not bromobenzene (C6H5Br).

6.  Line 168:  I think the reaction is with resorcinol and not phenol

7.  Line 178: again, this is a nucleophilic aromatic substitution process in which deprotonated 2-methoxyphenol is the nucleophile.

8.  Line 218: These are not an electrophilic aromatic substitution processes, they are metal-mediated coupling reactions.

9.  Line 309:  this book should be listed as a reference in the references in a standard format.

10.  Scheme 21:  An iodine atom is missing from the first two structures at the position para to the OH/OMe.  Also, the phenol reagent is not included in the scheme that would be required to form the product.

11. Line 74:  Yikes! The methoxy group is NOT turned into a proton!!  The bronsted or lewis acid instead complexes to the oxygen of the methoxy group, allowing nucleophiles to attack the methyl group and displace the phenol as a product.

12.  Scheme 26: it needs to be made clear that the initial structure is "Ar".  Perhaps under the first structure it can be written = ArMgBr

With the changes listed above I would support publication of this work.

Author Response

Dear Reviewer,

Thank you for taking your time to review our article on recent advances in the synthesis and applications of m-aryloxy phenols. We appreciate your constructive feedback, which we have fixed those issues in our revised manuscript. Following are fixes made in the paper:

  1. We have now included percent yields in some reaction schemes, as requested, except for schemes 7, 10, 15, 18, 19. Authors did not provide yields for these products in their papers.
  2. We have also provided the structure of resorcinol in the first reaction scheme (Scheme 4) in which it was mentioned.

3,7,8.   Thank you for pointing out our mistake in lines 121, 178, and 218. We have made the necessary corrections to our descriptions. We replaced citation for these reactions to better reflect description them.

  1. We made corrections of the typo on line 129
  2. We fixed Scheme 6 to show benzyl bromide (PhCH2Br), instead of bromobenzene (C6H5Br).
  3. Correction was made in line 168.
  4. In line 309, book reference was fixed.
  5. Scheme 21 was fixed.
  6. We appreciate your feedback on line 74 and have revised the text to more accurately reflect the process described.
  7. ArMgBr was added to Scheme 26

We hope that these changes address all of your concerns and that you will find our revised manuscript to be suitable for publication. Thank you again for your time and expertise in reviewing our work.

Sincerely,

Moldyr Dyusebaeva

Reviewer 3 Report

The manuscript is suitable for acceptance after the errors marked on the scanned cooy I am sending have been dealt with.

Author Response

Dear Reviewer,

Thank you for taking your time to review our manuscript and for providing valuable feedback. We appreciate your efforts and are grateful for the opportunity to revise our work.

Your suggestions for the changes was very valuable and well done.

We have made changes to the manuscript, including fixing the grammar and spelling mistake. We have also modified the format of author names to include only last names instead of initials.

Furthermore, thank you for pointing out at our mistake regarding nucleophilic process. We made correct changes and revised our references to better reflect this.

Once again, thank you for your valuable feedback and expertise. We hope that the changes made to the paper will address all of your concerns. We hope that our revised manuscript is now suitable for publication.

Best regards,

Moldyr Dyusebaeva